# "Mothers will be lucky if utmost receive a single scheduled postnatal home visit": An exploratory qualitative study, Northern Ethiopia

**Yemane Berhane Tesfau**[1,2]*, **Tesfay Gebregzabher Gebrehiwot**[2], **Hagos Godefay Debeb**[3], **Alemayehu Bayray Kahsay**[2]

**1** College of Medicine and Health Sciences, Adigrat University, Adigrat, Ethiopia, **2** Mekelle University, College of Health Sciences, School of Public Health, Mekelle, Ethiopia, **3** Tigray Regional Health Bureau, Mekelle, Ethiopia

* yemaneberhane12@gmail.com

**Data Availability Statement:** All relevant data are within the paper and its Supporting Information files. Raw data cannot be shared publicly because

## Abstract

### Background

Postnatal home visits (PNHVs) have been endorsed as strategy for delivery of postnatal care (PNC) to reduce newborn mortality and improve maternal outcomes. Despite the important role of the Health Extension Workers (HEWs) in improving the overall healthcare coverage, PNHV remains as a missed opportunity in rural Ethiopia. Thus, this study aimed to explore the barriers and facilitators of scheduled postnatal home visits in Northern Ethiopia.

### Methods

We conducted an exploratory qualitative study on a total of 16 in-depth interviews with HEWs and mothers who gave birth one year prior to the study. In addition, focus group discussions were conducted with HEWs and key informant interviews were conducted with women development group leaders, supervisors, and healthcare authorities from April to June 2019 in two rural districts of Northern Ethiopia. Discussions and interviews were audio recorded and transcribed verbatim in the local language (Tigrigna) and translated into English. The translated scripts were thematically coded using Atlas ti scientific software. Field notes were also taken during the discussion and while conducting the interviews.

### Results

Health system factors, community context, and individual level factors were considered as the barriers and facilitators of scheduled PNHVs. Leadership, governance, management, support and supervision, referral linkages, overwhelming workload, capacity building, logistics and supplies are the major sub-themes identified as health system factors. Physical characteristics like geographical location and topography, distance, and coverage of the catchment; and community support and participation like support from women's development groups (WDGs), awareness of the community on the presence of the service and

they contain potentially sensitive and identifiable information.

**Funding:** This study was carried out by Adigrat University and the " Tigray KMC Project" which is funded by Bill and Melinda Gates foundation and World Health Organization [Grant Number: 201526690]. The funding body had no role in the design of the study, collection, analysis, interpretation of data, and in writing the manuscript.

**Competing interests:** The authors have declared that no competing interests exist.

**Abbreviations:** ANC, Antenatal Care; CHW, Community Health Worker; CS, Cesarean Section; EDHS, Ethiopia Demographic Health Survey; FGD, Focus Group Discussion; HEP, Health Extension Package; HEW, Health Extension Worker; IDI, In-depth Interview; KII, Key Informant Interviews; KMC, Kangaroo Mother Care; PNHV, Postnatal Home Visit; PNC, Postnatal Care; UNICEF, United Nations Children's Fund; WDG, Women's Development Group; WHO, World Health Organization.

cultural and traditional beliefs were community contexts that affect PNHVs. Self-motivation to support and intrinsic job satisfaction were individual level factors that were considered as barriers and facilitators.

## Conclusion

The finding of this study suggested that the major barriers of postnatal home visits were poor attention of healthcare authorities of the government bodies, lack of effective supervision, poor functional linkages, inadequate logistics and supplies, unrealistic catchment area coverage, poor community participation and support, and lack of motivation of HEWs. Henceforth, to achieve the scheduled PNHV in rural Ethiopia, there should be strong political commitment and healthcare authorities should provide attention to postnatal care both at facility and home with a strong controlling system.

## Background

Postnatal care (PNC) is the most neglected aspect of maternal and newborn care in low and middle-income countries, despite it covers a critical transitional time. Studies in low and middle -income countries with high new-born mortality demonstrated that early postnatal home visits by community health workers help to reduce neonatal deaths and improve maternal and neonatal health [1].

Based on the experiences and evidence from South Asian trials, in 2009, World Health Organization (WHO) and United Nations Children's Fund (UNICEF) issued a joint statement recommending postnatal home visits (PNHVs) for delivery of postnatal care. Following the 2009 Joint Statement, many countries adopted policies to deliver postnatal home visits. Among the 75 countries included in the Countdown to 2015 report, 59 have policies to deliver such home visits within one week of birth [2, 3].

In Ethiopia, maternal and neonatal mortality remains the highest among the world, with 412/100,000 and 30/1,000, respectively. The trends of the Ethiopian demographic health surveys (EDHS) showed that there was a continuous decline in infant and under-5 child mortality preceding each respective survey. However, the trend among the neonatal mortality decreased from 39 to 29 between the 2005 and 2016 EDHS, but has remained stable since the 2016 EDHS [4].

Despite the fact that improvements observed on antenatal care (ANC) and delivery at health facility, PNC utilization at health facility remains low due to enormous reasons such as poor priority, unavailability, inaccessibility, poor quality of health services, socio-cultural beliefs, poor awareness of women on danger signs, not told to get the service, and distance [5–16]. Studies also showed that there are significant proportion of mothers prefer to return home or discharged within a few hours after delivery which makes them not to receive the required care [17, 18].

Postnatal care is one of the Health Extension Program components under the category of Family Health. The health extension program, which was launched in 2003 contributed to mobilizing community members towards the utilization of antenatal care and institutional delivery. HEWs were trained to promote utilization of basic maternal and child healthcare services which plays major role in improving the health of the mothers and newborns during the antenatal, delivery and postnatal period. HEWs are responsible to spend 75% of their time in the community and provide essential health services through the house to house visit [19].

Though HEWs are known in creating better health awareness among the community, hence, low coverage of postnatal visit within 48 hours was reported [20]. Besides, the coverage within three days after delivery from three Countries (Bangladesh, Nepal, and Malawi) by Community Health Workers (CHWs) showed that 57%, 50%, and 11% respectively. The pooled results of the study in these countries found that early visits were more likely if a mother had been visited by CHWs during pregnancy, birth notification by CHWs, and home deliveries [21].

The health system in many countries is strongly committed to and effective in reaching pregnant women with antenatal care services, but a similar commitment to postnatal care services does not yet exist. Evidences showed that lack of prioritization of postnatal care by government bodies, community health worker motivations, supportive supervision, training, logistics, workload, incentives, and community support were major factors that influence CHWs performance [22–26].

In Ethiopia, evidences show low coverage of PNHV by HEWs [27, 28]. Particularly, the coverage is extremely low in northern Ethiopia.i.e. 6.6%, 0.85%, 0.71% within 24 hours, three days, and seven days respectively [28]. The major factors associated with early PNHVs were HEWs visit home during pregnancy, skilled delivery, and having HEW's cell phone and no association were observed with maternal socio-demographic characteristics and early PNHVs [22, 29].

In Amhara and Southern Nations, Nationalities and Peoples of Ethiopia, a study was conducted focusing on barriers and facilitators of early PNHV coverage by CHWs. According to the study, the major barriers were physical factors like distance and transport, work issues like availability of HEWs and organizational work ethic; and information issues [30].

However, motivational factors at individual level, health system factors like supportive supervision, leadership and governance, and community context were not addressed in the study.

In northern Ethiopia, there is a dearth of evidence on why HEWs does not conduct PNHV. Hence, this study explored why rural mothers and their newborns in Northern Ethiopia do not receive postnatal home visits based on the recommended schedule from HEWs.

## Materials and methods

### Study design

This exploratory qualitative research employed thematic analysis as a methodological orientation to explore the barriers and facilitators of scheduled postnatal home visits.

### Participants

We included several participants to allow us multiple responses and explore the postnatal challenges. The study sample includes a convenient sample of HEWs, health center supervisors, and district level healthcare authorities. We also included WDG leaders, and delivered mothers. We identified, delivered mothers from HEW registers and were selected through a purposive sampling procedure considering whether the mother received PNHV, facility or home delivery and by the distance they lived from the health posts. The delivered mothers were approached through WDGs. A 1hours' time for one way was used. Besides, the participants were selected based on their ability to communicate and to express their opinion regarding the existing reality towards postnatal care.

Table 1 shows the characteristics of study subjects and methods of approach.

**Table 1. Methods of approach and participants.**

| Methods | Participants | No. |
|---|---|---|
| Focus group discussions (FGDs) | Health Extension Workers | 18 |
| In-depth interviews (IDIs) | Health Extension Workers | 8 |
| | Delivered mothers | 8 |
| Key informant interviews (KIIs) | Women Development Group Leaders | 4 |
| | Health center supervisors | 2 |
| | District level healthcare authorities | 2 |

## Study setting

The study was conducted in two rural districts of Tigray region, Northern Ethiopia. The two districts (Degua temben and Enderta) are found in south eastern zone of Tigray regional state. The zone serves a population in excess of 567,700 inhabitants with the total household estimated at 129,031. With regard to the number of health professionals, there were 731 health care providers in the zone out of which 183 were Health Extension Workers (HEWs) [31].

In rural Ethiopia, HEWs work in the health post which is found under the smallest administrative unit named as Tabia (sub-district). They were supervised by a health worker (preferably nurses or midwifes) who were assigned at health centers within the district. The HEWs are linked to their local community, to whom they serve and receive referrals. Antenatal care coverage and facility delivery of the zone was 97% and 89.2% respectively. Of all facility deliveries, only 18.2% of them had a minimum facility stay of 24 hours post-delivery. Postnatal home visit coverage by the HEWs within three days was 14.5% with only 0.71% mothers receiving the scheduled three postnatal home visits: within 24 hours, three days, and seven days [28].

## Data collection

Qualitative data were collected from April to June 2019. We conducted key informant interview, in depth interview, and focus group discussion with different participants (HEWs, Mothers, WDG leaders, Supervisors and district level healthcare authorities) to triangulate and validate the data. Eight HEWs participated in the IDI and lasts an average of 30 minutes. Initially, four FGDs (two FGDs from each of the districts) were planned. But due to data saturation three FGDs were conducted with 6 HEWs in each group which lasted an average of 60 minutes.

In general, the number of the participants was based on the saturation of the data (participants' descriptions become repetitive).i.e. we continued sampling the participants until no new information emerged and saturation was reached.

A total of eight mothers (one from each health post) were included in the IDI which lasts an average of 35 minutes. Four WDG leaders, two supervisors, and two district level healthcare authorities were included in the KIIs which lasted 25–60 minutes. For the health extension worker supervisors' we selected a convenience sample of one health center easily accessible for transportation from each district and interviewed each supervisor in the respective health centers.

The interviews for the district level healthcare authorities, supervisors and HEWs took place at their working place, while, the interviews for the delivered mothers and WDG leaders took place at their homes. However, the FGD for the HEWs was held in a public hall in the respective districts. The first author (YBT), and other two PhD candidates at the school of public health (ADW and GGG) who are well experienced and trained in qualitative research, conducted the interviews and FGDs. The guiding tools were prepared in English and translated

into the local language (Tigrigna). The pre-test were conducted before the commencement of the actual data collection. The guides were designed to elicit information about the experience of HEWs on home based postnatal care, barriers and facilitators, the role of HEWs and suggestions for improvement.

## Data analysis

The FGDs and interviews were audio recorded and transcribed verbatim in the local language (Tigrigna) and translated into English. Also field notes were taken during the discussion and interviews. Both the FGDs and interviews were carefully read through all the transcripts by the first author (YB). The transcribed data were then transferred to Atlas ti scientific software (version 7.5.4) for creating a cod and managing it. And the coding was completed by going line by line throughout the document. To check the inter-coder reliability, another author (TGG) coded all the FGDs and four interviews. Inductive and deductive coding was used to search additional coding throughout the transcriptions and themes. To verify our interpretation as sound, all the authors checked the document and tried to brief about the contents to a sample of the participants in the study districts.

## Ethical approval

Ethical approval was secured from the institutional review board of Mekelle University, college of health sciences (Reference number: 1437/2018). Besides, permission was obtained from the Tigray regional state health bureau and from the two district health offices. Also, informed consent was obtained from all participants. To ensure the participants' autonomy, they were informed about the objective, the risks, and benefits of the study. The right to privacy, and confidentiality were also considered throughout the study. Participants were also informed about their right to withdraw from the study at any time. Permission to record the FGD and an interview was also sought from the participants.

## Results

### Socio-demographic characteristics of the respondents

A total of 42 individuals (26 HEWs, 8 mothers, 02 district level healthcare authorities, 02 supervisors, and 04 women development group leaders) were participated in the study.

Overall, the mean age of the participants is 31.02 years (±6.82) with 71.4% married. From these, ninety five percent of the participants were female. More than three-fourth (77%) of the HEWs had reached level four in their education. Also, more than 42% of the HEWs lived outside their working tabias and their median working experience were 14 years with a range of 2 to 15 years. More than 92% of the HEWs had at least one child and 35% of them are divorced.

All the mothers (100%) participated in the interview conducted at least one ANC visit at the facility and 87.5% of them gave birth at the facility. However, 75% of the mothers had no experience in WDG participation. The median distance of mother's home to the health post takes 30 minutes with a minimum of 15 minutes and maximum of 75 minutes (Table 2).

### Identified themes

The major themes which emerged from the data included: 1) Health system factors, Community context, Individual factors, Consequences, and Suggestions (**Table 3**).

**Table 2. Characteristics of the study participants in the rural districts of Northern Ethiopia.**

| Total sample | HEWs | | Mother | Supervisors | District level healthcare authorities | WDG |
|---|---|---|---|---|---|---|
| | FGD (n = 18) | Interview (n = 8) | Interview (n = 8) | Interview (n = 2) | Interview (n = 2) | Interview (n = 4) |
| **District** | | | | | | |
| D.Tenben | 12 | 4 | 4 | 1 | 1 | 2 |
| Enderta | 6 | 4 | 4 | 1 | 1 | 2 |
| **Age** | | | | | | |
| 18–24 | 2 | 2 | 3 | 0 | 0 | 1 |
| 25–29 | 3 | 2 | 2 | 0 | 0 | 1 |
| 30–39 | 13 | 1 | 3 | 0 | 2 | 1 |
| 40–49 | 0 | 3 | | 2 | 0 | 1 |
| **Educational status** | | | | | | |
| No education | 0 | 0 | 3 | 0 | 0 | 2 |
| Primary | 0 | 0 | 1 | 0 | 0 | 2 |
| Secondary | 0 | 0 | 3 | 0 | 0 | 0 |
| > = Diploma | 18 | 8 | 1 | 2 | 2 | 0 |
| **Work experience** | | | | | | |
| <5years | 3 | 1 | NA | 1 | 2 | 2 |
| 5-15years | 15 | 7 | NA | 1 | 0 | 2 |
| **Place of delivery** | | | | | | |
| Facility | NA | NA | 7 | NA | NA | NA |
| Home | NA | NA | 1 | NA | NA | NA |
| **Distance to Health post (one way)** | | | | | | |
| <30' | NA | NA | 2 | NA | NA | NA |
| ≥30' | NA | NA | 6 | NA | NA | NA |

NA = Not Applicable.

## Health system factors

### Leadership, management, and governance

Almost all participants described that the government officials and leaders starting from the top level down to the implementers did not give attention to postnatal care compared to other

**Table 3. List of themes and sub-themes emerged from the data, rural districts of Northern Ethiopia.**

| Major themes | Sub-themes |
|---|---|
| Health system factors | Leadership, management, and governance |
| | Support and supervision |
| | Workload |
| | Perceived gaps on capacity building and incentives |
| | Perceived lack of supplies and logistics |
| | Linkage |
| Community context | Physical characteristics |
| | Community support and participation |
| Individual factors | Self-motivation to support the mothers and newborns |
| | Intrinsic job satisfaction |
| Consequences | Maternal death |
| Suggestions | Improve political commitment and controlling mechanisms, linkages, community participation |

programs. There is no uniform registration book/ log book for home based PNC services that HEWs can use and the supervisors also did not support the HEWs based on the recommended schedule. Erratic drug and other logistical supplies for the health posts indicate that a low priority is given to postnatal care services.

*"As to me, PNC did not get attention like other health services even from the top government authorities involved in the health system. For example, when individuals from the health system visit health post to take reports, their main concern is the number of mothers who completed fourth ANC and facility delivery. They do not ask us about home based postnatal care. They only compile the reports that they think as a core. Surprisingly, we do not have a uniform known registration book for documenting PNHV, rather what we do is document in a different piece of paper"* (**30–39 years old HEW, FGD**).

Another participant argues as follows: *'I could say the logistics and supplies available for postnatal care service in health facilities is not more than 15%. There is a huge gap on what the leaders said and the situation on the ground. If there is no necessary equipment like blood pressure apparatus (BP), thermometer, and newborn and maternal weighing scales, it is difficult to talk about HEWs are providing maternal and newborn health services. If a mother has not measured her BP or assessed her condition, how can she return for other services? It is also the same for home visits; the mother could not call for HEWs for PNHV. Except two, almost all health posts in our district have no BP apparatus. Currently, the function of HEWs and WDG leaders is almost similar. To consider a health post as health facility, it should be equipped with the basic materials'* (**30–39 years old healthcare authority, interview**).

Interviewed beneficiaries and service providers of the service also iterated this home based postnatal visit as almost non-existent or unscheduled if it exists.

*"This home based care, after delivery; the word itself seems for the sake of politics because it lacks attention by the concerned bodies. I have delivered three children, but no health extension worker visits my home after delivery"* (**30–39 years old mother, interview**).

*"I told you already that it is difficult to do this based on schedule even in a time when materials are available. Our focus is on the delivery, let say if the mother delivers at the facility. I am really happy and feel comfortable but if she delivers at home I will disappoint and become anger. If an HEW visits home after delivery utmost once, the mother will be lucky"* (**18–24 years old HEW, interview**).

The health extension worker supervisors also reported that the HEWs did not conduct PNHV while the district's priority is making home delivery free. They reported that let alone to provide postnatal home visit by HEWs, there were children that had never been vaccinated in their catchment.

*"The district's priority is to make home delivery free. If a woman delivers at home, we consider it as a death occurred. Thanks to God for keeping the mothers not to develop postpartum complications, otherwise, HEWs do not visit them. Let alone to conduct a postnatal home visit within seven days, wow, there are children, even who did not take a vaccine until one year"* (**40–49 years old HEWs' supervisor, interview**).

Participants expressed that PNHV is not conducted in a scheduled and planned way, because it is not clearly stated in the districts as a priority challenge. All HEWs, supervisors, district level healthcare authority and WDG leaders and some mothers assured that HEWs do

not hold necessary equipment during PNHVs when they conduct it. HEWs also explained that the supervisors and leaders do not motivate and provide them recognition based on their performance. Besides, they were not performing their duty according to what have been trained in the schools, because they do not have a clear job description. *The recognition and evaluation of HEWs were also undertaken based on the involvement of HEWs into politics.*

> *"For example, if we failed to attend a cabinet meeting that could be held during Saturday and Sunday, because we might be involved in providing vaccination and/food supplementation to the mothers and newborns, they will directly call to the district health office. And the district health office does not care about the maternal and newborn health and they ordered us to leave our duty and go to the cabinet meeting. If you let me talk about the current status of HEWs, we are doing other activities out of our job description"* (**30–39 years old HEW, FGD**).

> *"About 30% of the HEWs in this District reside outside the working kebele. This contributes to the obstacles we are talking about. There is a problem in conducting PNHV within 24 hours, three days and seven days. We always raise this issue in the cabinet meeting, but the district administrators do not bother about the postnatal care rather they focus on the politics. If we raise such questions, the district leaders assumed as if we are cheating them. The regional administrators also know this issue. By the way, expecting a positive response from the cabinet meeting is unthinkable. They only evaluate you if any maternal death occurred in the district"* (**30–39 years old healthcare authority, interview**).

The current governance system does not consider that HEWs have dual responsibility. All HEWs described that there have been a gap in HEWs development and staff transfer. There are also unclear guidelines on transfer of HEWs.

> *"As we are females, we get suffered a lot, fatigue and burnout. There is also a gap on governance. For example, if I need access to education, I should get the chance on time. However, I finished level four after 10 years. I have been 14 years since I completed level 3, but still I have not get a chance to learn a first degree because of the absence of clear and uniform performance evaluation across the districts. As a result, I will be disappointed because I do not think so that there is fair treatment of the employees. Another critical thing is the issues of living separately for a lifelong time from parents "*(**30–39 years old HEW, FGD**).

Most HEWs and all the WDG leaders narrated that the HEWs perform most of their activities especially PNHV by delegations.

> *"Since the health extension workers have confidence in our performance (women's development group) and believe that the delivered mothers can be assessed by WDGs, they do not visit home to home during the postnatal period. Most of the time HEWs performs their duty, especially care after delivery by calling us"* (**25–29 years old WDG, interview**).

## Imaginary and fault finding supportive supervision

Overall, the participants perceive that supportive supervision and support from health care providers working in the facility is important to improve HEWs performance in conducting home visits and builds the links between community and health system. However, it is irregular and nonexistent due to the less emphasis given to the postnatal care services.

"*Leaders, health workers and supervisors are not keen to support us even when we request them. Let alone to support us professionally using checklists, they are not responsive even when we request them for the essential drugs*" **(25–29 years old HEW, FGD).**

The participants also expressed the supportive supervision is undertaken in a fault finding way the fact that support and supervisions of health extension workers are expected to be conducted on a regular and supportive way.

**"***When supervisors come to take report, they look at faults and blame us. The supervisors do not also request us about postnatal care. I think it is not only due to lack of knowledge that the supervisors were not providing support, but also due to a poor commitment of the supervisors and they thought community based maternal and newborn care is the duty of HEWs*" **(30–39 years old HEW, FGD).**

Interviewed HEWs also reported this as: "*When higher officials from the regional health bureau, district level health extension program coordinators and supervisors from health centers come and observe our performance, we become happy. We will be motivated by the good performance we accomplish, but our morale will be hurt if they do not recognize our work*" (**40–49 years old HEW, interview**).

Due to lack of regular and relevant assignment of trained supervisors and poor supportive supervision, health extension workers were not doing postnatal care as needed.

"*Especially, recently we have not done this (PNHV) due to poor support we received from the WDG and from the supervisors. We do not have supervisor in this Tabia. They assigned one supervisor today and leave for another day. In this Tabia we serve as HEWs as well as Supervisor. No one supervised me. I have been alone in this tabia for more than two years.*" (**30–39 years old HEW, interview**).

The supervisors interviewed also reiterated these feelings, by saying the following:

**"***I am actually dissatisfied being supervisor because I am working as provisional supervisor. They have not provided me an opportunity to be full supervisor. I can say we are not doing our assignment. We are not supervising all HEWs. There are also false reports because there are no technologies to check this. We only receive the reports for our consumption because the higher government bodies depend only on report*" (**40–49 years old HEWs' supervisor, interview**).

HEWs' supervisors that assigned in the health centers lack adequate knowledge and skills to support and they have the perception that HEWs can accomplish by themselves.

"*I did not receive training on supportive supervision, especially PNHV, which I have difficulties. I do not know this (if a mother delivered at facility stays for at least 24 hours) because I am a newly assigned supervisor. But, from my observation mothers are discharged early in this facility. For example, two mothers have delivered and discharged immediately after four hours. From our side, the care providers do not support to the health extension workers as needed because most health care providers have the perception of being a clinician. They consider Preventive services, community mobilization and maternal and newborn care at home out of their scope*"(**40–49 years old HEWs' supervisor, interview).**

## Overwhelming workload of HEWs

Almost all the HEWs and other participants considered the duty that is assigned to HEWs is overwhelming. And they considered as a barrier to perform postnatal home visits in a scheduled way. The study participants expressed that HEWs are performing multiple tasks such as mobilizing the community to enroll in the CBHI, immunization campaigns, environmental sanitation campaigns, distributing nutritional supplements, conducting conferences with WDG, growth monitoring for children, making maternal and newborn referrals to health centers, mobilizing the rural community to utilize fertilizers and providing health education. The participants also emphasized that the number of HEWs and the total population is incomparable that may hinder the performance of HEWs to conduct PNHV.

*"Let alone to conduct a visit three or four times, it is even very difficult to make one visit with a standard schedule. For example, if today is a vaccination day, I could not make a visit because it is difficult to cover all activities with one or two HEW"* (**30–39 years old HEW, FGD**).

*"For example, recently there were about three to four facility deliveries in this catchment but I did not visit them, because I do have many assignments like mobilizing the community for health insurance, immunization, sanitation campaigns, and money earnings etc. We, even visit only one times for those mothers who develop complications. I can say there is no work that HEWs are not involved"* (**40–49 years old HEW, interview**).

The HEWs reported that the number of HEWs assigned in the Tabia and the population in the catchment is not proportional. They also rose HEWs have extra burden and because of workload, the HEWs delegate their role to WDG leaders. However, some participants stated that the number of HEWs did not matter for effective performance.

*"With respect to coverage since there is above 10,000 population in this catchment, it is difficult to cover all delivered mothers with two HEWs"* (**30–39 years old HEW, FGD**).

*"There is also a difference in performance of HEWs from catchment to catchment. It is not the number of HEWs that matters for effective performance, because there is a catchment with only one HEW but performs well as compared with those catchments that have two to three HEWs"* (**40–49 years old HEWs' supervisor, interview**).

## Perceived gaps on capacity building and incentives

The participants mentioned gaps in their training, both short terms, refresher, and long term trainings/continuing education. However, except some of the HEWs who have the short working experience, they have no problem with the knowledge and skills of postnatal care. Lack of incentives for undertaking postnatal home visits especially those distant once are considered as major challenges by the HEWs. In general, almost all health extension workers considered training and home for residence as an incentive. Unfair treatment of HEWs, especially, for the long term training/continuing education affects their performance.

*"Except for those newly employed HEWs, providing PNHV is not difficult for those who serve more than three years. We have enough knowledge"* (**30–39 years old HEW, FGD**).

*"I have been living here but I have no house of my own. I need a house because the training does not change my life. In the past, the regional health bureau sent us to take an entrance exam, but only 40 HEWs got the chance out of the 280 HEW participated in the exams. You*

*can understand from this the regional health bureau has no interest to let us upgrade our status. There is also a problem in selection of HEWs to participate in entrance exam because, the evaluation criteria are subjective. When we see such bureaucracy, we disappointed and become hopeless and even think to shift our role to become a merchant "(**30–39 years old HEW, FGD**).*

*'We know as government employee, we have salary. However, it would be good if they provide us incentives like for a gown, compensation for house rent. It is also good if they provide us money for duty. They do not provide us any incentive except a 100 Birr mobile card' (**30–39 years old HEW, FGD**).*

*"We evaluated that the postnatal care at home in the district is poor. Thus, the motive of HEWs is also not as before probably because of the work load they have" (**30–39 years old healthcare authority, interview**).*

### Perceived lack of supplies and logistics

The participants enlighten that almost all health extension workers do not hold necessary equipment and other supplies except some drugs like TTC, and Iron foliate when they conduct home visits. They were unable to render services to the mothers and their newborns because of lack of essential supplies or nonfunctional equipment, such as BP apparatus, and thermometer.

*"We have tried to conduct PNHV but we do not have BP apparatus. We have requested to health centers but they do not have even for themselves. I stayed here for the last two years, but I have not seen any BP apparatus and I did not check mothers' BP let alone at home, even at the health post"(**18–24 years old HEW, FGD**).*

*"They do not have necessary materials when they make a postnatal visit. I have never seen an HEW visiting home to home using BP apparatus to help the mothers" (**18–24 years old WDG, interview**).*

*"I stayed in this tabia for the last three years, but I have never seen any BP apparatus in the clinic. When we request the health center, they do not provide us a positive response and they told us it was in the procurement. Also, there is no clear registration or log book for documenting PNHV" (**330–39 years old HEW, FGD**).*

*"Shortage of equipment might be another issue; because I have never seen HEW that visit home using B/P and Thermometer in this area"(**18–24 years old mother, interview**).*

Interviewed participants also noted that a lack of supplies; such as checklists and reporting log books for home based postnatal care.

*"We have not given an emphasis about the necessary equipment. We do not have clear log book for PNHV, what components provided, we do not have a postnatal checklist" (**40–49 years old HEWs' supervisor, interview**).*

*"Except some health posts, they do not have BP apparatus. Even if they have it might be nonfunctional. There are also other issues, whether really the HEWs are visiting home using BP. They are reluctant to use BP at every visit. Therefore, from my opinion, it is difficult to believe HEWs do home visit using the necessary equipment like BP apparatus and thermometers during the postnatal period due to this the mothers become reluctant to call them rather they*

*choose to visit health facilities if they develop complications"* (**30–39 years old healthcare authority, interview**).

## Linkages

In general, the participants highlighted that getting information about the condition of the mother and/the newborn is essential for the HEWs to conduct PNHV. Despite this, however, almost all participants expressed that this postnatal home visit isn't based on the standard schedule and it is not planned when it exists. The presence of functional linkages (from facility and community) either by calling through phones, or by a means of contact in person by WDG leaders, place of delivery, traditional seclusion, and the presence of complication were the main issues raised as information factors.

*"On the case of facility delivery, since we assume the mothers stay in the facility for the first 24 hours, we focus for the rest visits if the health care providers inform us in a condition when the mother could develop complication or for those home deliveries. However, it is not planned. We do not do home visit intentionally for the purpose of postnatal care. We conduct the visit after ten days because of delay in information about the delivery of the mother"* (**25–29 years old HEW, FGD**).

In the study districts the Health extension program coordinators also described postnatal home visit is not running in a scheduled way because of the presence of poor linkage between the facility and health extension workers.

*"The families of the delivered mothers come to a facility by traveling two-three hours while they do not have enough money to buy any food. They bring dry bread from their home: You can imagine how a delivered mother could eat this dry bread. Thus, they need to be discharged immediately that makes difficult for the HEW to get adequate information about the status of the mother because HEWs' expectation is to stay the mother at least 24 hours at the facility. When the mother is discharged from facility, there is information gap b/n the HEW and the mother"* (**30–39 years old healthcare authority, interview**).

The health extension workers dictated that complete information during pregnancy is helpful for the provision of a continuum of care for the mother and newborns. However, the information flow is not complete.

*"We may have information about some of the mothers during pregnancy. Sometimes health care providers could inform us about the condition of the mother, especially if the mother has complications. For those mothers delivered at home, however, we do not hear early"* (**30–39 years old HEW, FGD**).

This is also complemented with the notion that health extension workers did not also considered postnatal home visits for all mothers. This program is, however, stated not functional even in the presence of adequate information:

*"The health centers and hospitals do not give a call to HEWs. They do not even get information about the delivery of the mother. From my perspective before and within 24 hours after delivery, they give me good care, but after 24 hours no one saw me"* (**330–39 years old mother, interview**).

*"Most community members are reluctant to tell the HEWs about the delivery of the mother. They tried to hide be it facility delivery or home delivery. Surprisingly, there is a time where we hear about the delivery of the mother during baptism or at the church. It is known that most mothers are reluctant to call HEWs especially before making "**Gelleb**" a ceremony that could be made at the age of seven days even those whom we consider educated. They do not want to expose their newborn. For example, there are three HEWs in the health post, but they did not get information about the birth of the mother because the families tried to hide the information due to traditional beliefs and for seclusion"* (**30–39 years old healthcare authority, interview**).

## Community context

**Physical characteristics.** All HEWs mentioned that there are remote households that are inaccessible that faced several challenges due to terrain issues that predispose them to physical and sexual violence. The participants also raised about large coverage areas with limited transportation options. They stated that *"the coverage of the population is high which is not balanced with the number of HEWs and the area is a terrain with hills topographically. There are also remote households which are difficult to visit them alone. The standard is one HEW to 500 households, however, practically, more than 2000 households are covered by one HEW only"* (**25–29 years old HEW, FGD**).

*"We conduct PNHVs, but it is not based on schedule and standard. Especially, because of terrain geographical location, the coverage and quality of PNHV is low. I am alone here in the tabia and when I plan to visit to distant households, I need another person who supports me. However, the person whom to with me needs to arrange convenient day, because of this the schedule for PNHV will be affected"* (**30–39 years old HEW, FGD**).

*"For example, I have encountered one person attempting to harass me, but at that time two passengers had been on their way to other areas and saved me. There are also students whom they throw a stone. Therefore, visiting alone is difficult for distant households. Look, I am old, almost above 40 years, but a person whose age might be not more than 20 years attempted sexually to harass me. The young boy is almost at the age of my son, but he did that"* (**40–49 years old HEW, interview**).

All interviewed WDG leaders also described that almost all HEWs have not conducted PNHV for those distant households and even for those nearby households in the case of multiple deliveries.

*"The HEWs do not conduct PNHV in our village because the village where I am living is far from the health post, but what they do is they call and inform us to conduct the visit. For example, I am WDG leader and I delivered in hospital, but the HEW did not visit me because my home is very far from the health post"* (**25–29 years old WDG, interview**).

However, HEWs have not conducted postnatal home visiting even for those nearby households.

*"From my experience, I could not believe if HEWs said we have visited three times based on the standard schedule even for these nearby households"* (**25–29 years old HEW, interview**).

Other participants also raised this issue that mothers and newborns have not received PNHV from HEWs.

*"It is really difficult for the HEW to visit the mother at distant households travelling three and half hours for a trip, but there are nearby mothers at three to five minutes who do not receive the service"* (**30–39 years old healthcare authority, interview**).

**Community support and participation.**    Almost all participants in the study districts raised the support from the community, especially from the WDG leaders and the one to five networks after delivery is poor. The main reasons raised were the participation of WDG leaders in maternal and newborn care in general and postnatal care in particular is poor due to the dependency developed on NGOs. The WDG leaders were also reluctant for this service due to the poor emphasis given to this service by the government officials. The mothers and WDG leaders also do not want to report home deliveries because of fear of the HEWs and the government leaders. The awareness of the community on the benefits of postnatal care was also poor. Socio-cultural belief and practices also influence seeking of the service from HEWs.

Almost all HEWs and most other respondents reported that community support and participation by the WDG is deteriorating from time to time due to the dependency on incentives.

*"Currently, it is difficult to work with WDG leaders because they (WDG) do not want to be volunteers. When we inform them (supervisors) that WDG leaders are not working and are not supporting us, they push us to convince them. However, the top leaders report as if the WDG leaders are functioning well. They also raise the need to care their family"* (**25–29 years old HEW, FGD**).

*"There is no any payment for WDG leaders and do not want to work for free. Because of this their interest is poor. Despite the availability of information, they do not care about informing HEWs on time. They are unwilling on serving the community. Their intension is generating income and keeping for incentives. For example, they say does the government lack to pay 50 Birr per year"* (**30–39 years old HEW, FGD**).

*"Currently, the WDG leaders do not support us, because, of the competing interests. They have a dual responsibility; to participate in agricultural and domestic responsibilities. If we ask them to help to mobilize community and visit the households, they respond to us: You are working because you have salary they respond to us"* (**40–49 years old HEWs' supervisor, interview**).

*We WDG leaders also decreased in connecting the mothers with HEWs due to there are no incentives even for cards to call for HEWs* (**40–49 years old WDG, interview**).

However, interviewed participants contradicted to the idea that there is a poor community support: it is, because the HEWs themselves did not provide a due respect for the mothers, and the community is having not well aware about the presence of the service.

*"As a chance I delivered this year during January in Ayder Hospital when there was no service provider in the nearby hospital. But after I returned home no one visit me for postnatal care. I myself dislike them due to the poor respection provide for us. Second the communities do not know the presence of the service. They do not visit me at home even when I delivered through CS"* (**30–39 years old mother, interview**).

"*Awareness among the community is poor. Let say a husband doesn't worry about the postnatal care once his wife delivers safely and he prays to the god for delivering safely. There are also community members who want traditional seclusions because of spiritual beliefs like* **Deftera**" (**30–39 years old HEW, FGD**).

## Individual factors

**Self-motivation to support the mothers and newborns.** HEWs expressed their commitment to support the community; however, the postnatal home visit was not supported by the supervisors.

*"I will be happy and mentally satisfied when the community receives care at home"* (**30–39 years old HEW, FGD**).

*"I prefer, to visit home to assigning at the health post because you can have an opportunity to communicate many issues with family at home and you become satisfied after time when you observe many progress"* (**40–49 years old HEW, interview**).

*"I want the mother not to die because of my negligence. I become satisfied when the mother becomes healthy and productive; otherwise the salary we receive is not enough"* (**18–24 years old HEW, interview**).

*"Frankly speaking, HEWs are positive in solving the communities' health problem if we inform them about any problem in the community"* (**25–29 years old mother, interview**).

However, some of the participants stated that the motivation of health extension workers to support the community is poor.

*"They do not accomplish their responsibilities to the community. Due to that the communities do not trust them. They made a report, while sitting in their home as if they were providing the service. Let alone to come for the home visit, they do not want to pick their phones. They switch off their phones. It is good, even if they visit the nearby mothers if they were committed. In the rural community mothers have a strong feeling of fear to HEWs associated with the Fafa provision"* (**30–39 years old mother, interview**).

**Intrinsic job satisfaction.** Almost all HEWS in the FGD and IDI mentioned that they have no satisfaction being HEWs. One interviewee argues that *"one thing always disturbing me is, living separately for a lifelong from my kids and husband. I do not care about incentives rather what I am thinking is about the transfer from rural to urban areas or like other civil servant to arrange the career structure so that we can have an access to urban communities or change our job. If there are no such opportunities, you will be burnout and do not function well"* (**30–39 years old HEW, FGD**).

*"I have employed as HEW, but it was not on my knowledge when I joined this occupation throughout my stay for the last 14 years, nobody supports me. However, other individuals, who employed later with another profession upgraded and transferred. This is unfair, but I am here for the last 14 years. We (HEWs), most of us needs to change our occupation, however,* our *salary is not enough for doing other education. Because of this, the services we provide to the community are deteriorating from time to time. I do not know the reason why HEW is not assigned at health facilities and woredas. It is also one means of divorce because if*

*my husband transferred to other places /town we will be divorced automatically. Currently, I am comfortable if they allowed me to retire to give care for my family"* (**30–39 years old HEW, FGD**).

*"It is very stressful to be assigned as HEW; especially this year I have decided to leave this job because CBHI is annoying"* (**25–29 years old HEW, interview**).

One HEW, however, expressed that she has an intrinsic job satisfaction:

*"I am not interested in incentives. I do not consider even as an issue at any time. What motivates me is the recognition I received from the community. I have awarded certificates and even money by the women's affairs office, by the district's administrative office and regional authorities. Honestly speaking, it does not change me rather what impressed me is the recognition I received by the community"* (**40–49 years old HEW, interview**).

## Consequences

The participants mentioned that, both maternal and newborn complications have occurred. Most HEWs described that mothers are dying after delivery even when they delivered at the facility.

*"For example, about four mothers died in our catchment and all deaths occurred during the postnatal period. In addition to the four deaths occurred, one mother also died in this Woreda due to postnatal complications. I personally know the mother and she had been living around 10 minutes far from the health facility. There were three HEWs in the health post, but they did not get information about the birth of the mother because the families tried to hide the information due to traditional beliefs and fear of sanctions due to home delivery. The family believed that if the mother had taken to hospital they might inject her drug so that she would die of that drug because they strongly depend on traditional believes like believing in evil spirits"* (**30–39 years old Healthcare authority, interview**).

*"We are also alert for postpartum bleeding because here in our Tabia two-three mothers had passed away at their home due to postpartum bleeding"* (**40–49 years old HEW, interview**).

More than 50% of the mothers participated in the interview raised that complications on the newborn and mother and harmful traditional practices were observed in the community.

*"I did not receive care after delivery. I provide malted barley and butter for the newborn within seven days. And we enjoy by saying **elil**. . . i.e. expressing extreme happiness"* (**25–29 years old, mother, interview**).

*"I have seen many newborns suffering from cord infection. Some mothers apply oil and /butter to facilitate healing, but become complicated"* (**30–39 years old mother, interview**).

## Suggestions

Almost all the participants suggest that government bodies should give attention as apriority agenda for this service. Linkages among Health facility care providers, HEWs, and WDG leaders should be strengthened. HEWs need to develop commitment in accomplishing their duty. Sustained supplying of logistics and controlling of the program also needs consideration.

*"The government bodies starting from the region should give emphasis for community based maternal and newborn care especially after delivery. The supportive supervision needs scheduled and needs monitoring. Women Development Groups also need to work with HEWs. Awareness creation about the presence of the service at home needs consideration. The districts should consider fair treatments of HEWs and WDG leaders during selection for trainings"* (**30–39 years old HEW, FGD)**.

*"We HEWs are not all the same; for example some HEWs can say I am at house to house while they are seat at their home. Therefore the supervisors should use any checking mechanisms whether the HEWs are really doing their duty or not, but they do not do this currently. They only collect reports which might be false report"* (**40–49 years old HEW, interview**).

*"From my opinion the number of HEWs needs to increase. The shift of the roles of HEWs to politics should also considered"* (**30–39 years old HEW, FGD**).

## Discussion

Early Postnatal home visits (PNHVs) with high coverage by community health workers have been endorsed as a strategy for delivery of postnatal care (PNC) to reduce newborn mortality as well as to improve maternal outcomes. This period is also a time of transition when many women initiate new behaviors. The Ethiopian national policy document stated that HEWs can carry out this role within 24 hours, 3[rd] and 7[th] days. However, the opportunity to provide postnatal care with in the recommended schedule during this critical period is often missed in Northern Ethiopia [28]. Hence, this study explored why PNHV were not provided based on the recommended time schedule in Northern Ethiopia.

Health system factors (leadership, management and governance, support and supervision, workload, supplies and logistics, and information), community context (physical characteristics, and community support and participation), and individual factors (self-motivation to support the mothers and newborns, and intrinsic job satisfaction) were identified themes that are perceived as barriers and facilitators of scheduled PNHVs by the participants.

Even when CHWs possess the necessary knowledge and skills to provide services to their communities, they were often challenged by the health system of which they are a part, or the context in which they work [25]. Our finding showed that maternal and newborn health is theoretically considered the key health indicators at a policy level and almost all the participants recognized as a critical program. However, the leadership and coordination from the concerned bodies for postnatal care lacks attention practically. Evidences also showed that lack of prioritization of postnatal care by healthcare authorities, loose leadership support and lack of local government ownership and lack of accountability for the health extension program are indicators of poor attention for the program by the government, which in turn created poor performance among the HEWs [32–34].

In Ethiopia, lack of coordination between vertical programs and between various NGOs were reported, reducing the time health extension workers could spend in their communities [35]. Other studies also showed the recognition and integration of CHWs in the health system seem to be more important for CHW performance than the existence of a CHW policy [36, 37]. All participated HEWs, supervisors, district level healthcare authorities and most mothers and WDGs expressed that less emphasis is given to postnatal home visits even from the top leaders. They give priority to other services like 4[th] ANC and facility delivery; as an indication for this were almost all health posts were not equipped with postnatal drugs and supplies and almost all HEWs did not use any equipment while doing PNHV in case and no uniform and standard registration formats for the reporting of PNHVs. There were no clear performance

evaluation criteria based on which HEWs could be academically upgraded, rather they were evaluated based on their involvement in politics affiliated to the ruling party.

Almost all participants stated that HEWs were involved in other activities that are out of their job description like attending cabinet meetings. Issues with involvement in politics were also reported, especially in weak political systems, community health workers may be "rewarded" appointments for political support by local governments and politicians [38, 39].

These unfair treatments will raise the issue of poor governance. As a result the motives of HEWs were affected that leads to poor performance and burnout. Findings in other African countries also showed that programs that have low priority could result poor performance of CHWs [40, 41].

Overall, the participants viewed that supportive supervision and support from health care providers working in the facility is important and key to the success of HEP to build the links between community and health system. Evidences also show that the success of CHW programs hinges on regular and reliable support and supervision [40, 42–44]. However, in this study; it is irregular and non-existent due to the less emphasis given to the postnatal care services both at home and facility. Even when it did occur, the supervisors do not give emphasis for PNHVs rather they focus on other maternal health services. PNHV is not conducted by HEWs in a scheduled and in the recommended way; however, the leaders in the respective levels were collecting false reports as if the service is provided. A study conducted in rural districts of Ethiopia also reports that false reports were provided to the government bodies because of lack of scheduled and standard supervision [32]. The supervisors also raised an issue that they feel as if incompetent to provide technical support to HEWs, because they did not receive training on supervision and they were provisionally assigned. It is known that the support and Supervision of health extension workers are expected to be conducted on a regular and supportive way. However, the participants expressed that it is undertaken in a fault finding way. Besides, health workers in the facilities also considered community work as out of their scope. There were even health posts who do not have assigned supervisors. Qualitative findings on supervision of community health workers in Mozambique also demonstrated that the link between CHWs and supervisors were poor due to poor training, and lack of skills [40, 45].

Besides to the absence of support from health care providers and supervisors in the catchment, the duty that is assigned to HEWs is overwhelming. They considered as a barrier to perform postnatal home visits in a scheduled way, because currently HEWs are performing multiple tasks like mobilizing the community to enroll in the community based health insurance (CBHI), immunization campaigns, environmental sanitation campaigns, distributing nutritional supplements, conducting conferences with WDG and mobilizing the rural community to utilize fertilizers and providing health education. A systematic review in low and middle-income countries revealed that high workload was reported by CHWs. And this could result in lower motivation and ultimately lower performance [42]. The number of HEWs and the total population is not comparable because the structuring of the kebelle where HEWs were assigned was administratively demarcated before 30 years; because of this almost all HEWs did their role by delegations to WDGs. Let alone to conduct three scheduled visits within a week, it is difficult even to conduct a single PNHV as more than 42% of the HEWs lived outside their working tabias. Studies in other parts of Ethiopia and other African countries such as Lesotho, Malawi, and Bangladesh also revealed that lack of attention to working conditions and human resources management and the workload that can be described by the interplay of the number and organization of tasks and the catchment area could be factors to the low performance of CHWs [41, 46, 47].

In general, information about the delivery of the mother was important for HEWs to conduct PNHV even though that visit may not be scheduled and lacks revisiting. Almost all

HEWs were also reluctant for this service because of the attention given by the leaders in the district. And what they did is just they report any delivered mother if they encountered while doing community meetings or other activities as if they provide PNHVs. They did not visit for the purpose of postnatal care intentionally. There was no formally written document whereby WDGs could handover the delivered mothers, even when the WDGs call them about the delivery of the mother, what they did were just informing them (WDGs) to conduct an assessment by themselves. Also there were no clear linkage between the health facilities and health posts. Some health care providers, however, notify the HEWs through phones if they suspect any potential complications on the mother or her newborn. Even though an increase in facility delivery coverage exists, facility births were less likely to receive early postnatal care at home due to the communication gaps and presence of early discharge i.e. only 18.2% delivered mothers stay at facility for the minimum of 24 hours [28].

However, the HEWs were curious if the family members give them a call to visit them during the postnatal period, even though there were community members hide information due to traditional and cultural reasons, and lack of awareness about the service. Families of home delivered mothers' were also reluctant to call to WDG or HEWs because of fear of actual or presumed discrimination by HEWs and other government bodies. Home delivery may be a barrier for notifying HEWs for fear of sanctions because of fear of disrespect and abuse according to other studies done in Ethiopia [32]. Studies on postnatal experiences and barriers to utilization in Uganda and Zambia also showed that newborns delivered at home had not received postnatal care because of disrespect observed by the care providers which was a double jeopardy for the newborns [48].

Evidences showed that having enough job aids, equipment, and other drug supplies were often reported to facilitate CHW performance [49–51]. In this study almost all health posts did not have uniform registration book for PNHVs, necessary equipment like blood pressure apparatus and thermometer. The HEWs and the WDG leaders were doing similar function during postnatal home visit and the mothers were reluctant to call the HEWs because the HEWs do nothing both for the mother and newborns. However, findings in Nepal showed that the performance of CHWs have increased if they felt they had something of value to offer families like drugs that could increase the demand of home visits [22].

Absence of educational opportunities and lack of incentives were considered major factors for poor performance of CHWs [52, 53]. Almost all health extension workers considered training and home for residence as an incentive. Unfair treatment of HEWs especially for the long term training/continuing education affects their performance. Together with poor living, working conditions, and unequal treatment of HEWs, the above mentioned factors could negatively affect the performance of HEWs.

Physical contexts such as distance, and topographically inaccessible areas as a barrier to CHWs home visit have been reported in different studies [25]. In this study, distant households were not a concern for HEWs to conduct PNHV. However, geographically terrain areas were difficult for HEWs to conduct PNHV due to fear of physical and sexual violence. Similar findings were reported from a study conducted in New Guinea, where CHWs felt unsafe and scared because of substance abuse among young men, violent assaults, verbal abuse, and accusations [54].

Community participation and support as part of community context can equally affect CHWs performance on home visit with the physical characteristics and are much more challenging for CHW program managers to control. Yet, they are the most critical aspects for improving performance of CHWs [55, 56]. Our finding shows that poor WDG support and their participation has stalled or is reversing due to the need for incentives because they have double responsibility to support their family; and fear of blame for reports of home deliveries

by the family members and by the district leaders were observed. Family members hide about the delivery of the mother due to probably fear of denial of health and social services. For instance, a study in Zambia revealed that perceived fear of punishment and financial penalties were observed on home delivered mothers [48]. Cultural practices, beliefs, and lack of awareness on the availability of PNHVs by the family members were also a challenge. Similarly, studies in Africa and Asia showed that beliefs and awareness of the community challenges CHWs performance [37, 57].

Apart from the health system and community factors, individual motivations of CHWs affect their performance [32]. Most HEWS in the study demonstrated that they have no satisfaction being HEWs. More than 92% of HEWs expressed poor intrinsic job satisfaction being deployed as HEW. Studies else were show that socio-demographic factors especially marital status have mixed effects on community health workers performance [42]. In this study, HEWs expressed that this Health Extension Package (HEP) predispose them to divorce, because most of the husbands have high mobility to urban areas and they let them live together because of this they prefer to leave the program.

## Study limitations and strengths

Reporting bias from some of the participants especially from WDG is expected as they were selected in consultation to HEWs in each catchment. There could be also social desirability bias from the mothers, as all of the interviewers were males. It is obvious that there could be difficulty to transferability of the findings to other communities. However, we tried to access the experiences and views of different participants to maximize data quality and transferability on the barriers and facilitators of PNHVs with varying data collection methods. Open discussion and regular meetings was also conducted among the researchers to enhance reflective thinking and to maintain trustworthiness.

## Conclusions

Our finding suggests that the major barriers and facilitators of postnatal home visits based on the recommended time schedule from HEWs were poor attention of healthcare authorities, lack of effective supervision, poor functional linkages, inadequate logistics and supplies, unrealistic catchment area coverage, poor community participation and support, and lack of motivation of HEWs. To achieve the scheduled PNHV in rural Ethiopia, there should be strong political commitment and healthcare authorities should provide attention to postnatal care both at facility and home with a strong monitoring and evaluation system. Besides to the availability of functional linkages among health facilities, HEWs, WDGs and families, innovative interventions should be implemented to establish functional scheduled PNHV by fully harnessing the potentials of HEWs.

## Supporting information

**S1 File. FGD.**
(DOCX)

**S2 File. IDI Guide HEWs.**
(DOCX)

**S3 File. IDI guide mothers.**
(DOCX)

**S4 File. KII experts and supervisors.**
(DOCX)

**S5 File. KIIs guide WDG.**
(DOCX)

## Acknowledgments

The authors are grateful to the Tigray regional health bureau and the two district health offices for their facilitation and support of the fieldwork and to all respondents for their participation in this research.

## Author Contributions

**Conceptualization:** Yemane Berhane Tesfau, Tesfay Gebregzabher Gebrehiwot, Hagos Godefay Debeb, Alemayehu Bayray Kahsay.

**Data curation:** Yemane Berhane Tesfau, Tesfay Gebregzabher Gebrehiwot, Hagos Godefay Debeb, Alemayehu Bayray Kahsay.

**Formal analysis:** Yemane Berhane Tesfau, Tesfay Gebregzabher Gebrehiwot, Alemayehu Bayray Kahsay.

**Funding acquisition:** Hagos Godefay Debeb.

**Investigation:** Yemane Berhane Tesfau, Tesfay Gebregzabher Gebrehiwot, Hagos Godefay Debeb, Alemayehu Bayray Kahsay.

**Methodology:** Yemane Berhane Tesfau, Tesfay Gebregzabher Gebrehiwot, Alemayehu Bayray Kahsay.

**Project administration:** Yemane Berhane Tesfau, Tesfay Gebregzabher Gebrehiwot, Alemayehu Bayray Kahsay.

**Resources:** Tesfay Gebregzabher Gebrehiwot, Hagos Godefay Debeb, Alemayehu Bayray Kahsay.

**Software:** Yemane Berhane Tesfau, Tesfay Gebregzabher Gebrehiwot.

**Supervision:** Yemane Berhane Tesfau, Tesfay Gebregzabher Gebrehiwot, Hagos Godefay Debeb, Alemayehu Bayray Kahsay.

**Validation:** Yemane Berhane Tesfau, Alemayehu Bayray Kahsay.

**Visualization:** Yemane Berhane Tesfau, Alemayehu Bayray Kahsay.

**Writing – original draft:** Yemane Berhane Tesfau, Tesfay Gebregzabher Gebrehiwot, Alemayehu Bayray Kahsay.

**Writing – review & editing:** Yemane Berhane Tesfau, Tesfay Gebregzabher Gebrehiwot, Hagos Godefay Debeb, Alemayehu Bayray Kahsay.

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
