## [Decision Letter · Decision Letter 0]

25 Nov 2020

PONE-D-20-24965

“A mother will be lucky if utmost receives a single scheduled postnatal home visit”: an exploratory qualitative study, Northern Ethiopia

PLOS ONE

Dear Dr. Tesfau,

Thank you for submitting your manuscript to PLOS ONE. After careful consideration, we feel that it has merit but does not fully meet PLOS ONE’s publication criteria as it currently stands. Therefore, we invite you to submit a revised version of the manuscript that addresses the points raised during the review process.

The manuscript has been evaluated by two reviewers, and their comments are available below.

The reviewers have raised a number of concerns that need attention. In particular, they have suggested to follow the COREQ guidelines for qualitative reporting on the methodological aspect of the study. Finally, the language of the manuscript should be in an intelligible fashion and written in clear, correct, and unambiguous English. 

Could you please revise the manuscript to carefully address the concerns raised?

We look forward to receiving your revised manuscript.

Kind regards,

Lucinda Shen, MSc

Staff Editor 

PLOS ONE

Journal Requirements:

3. When reporting the results of qualitative research, we suggest consulting the COREQ guidelines: http://intqhc.oxfordjournals.org/content/19/6/349.

In this case, please consider including more information on the number of interviewers, their training and characteristics; and please provide the interview guide used.

Reviewers' comments:

Reviewer's Responses to Questions

**Comments to the Author**

1. Is the manuscript technically sound, and do the data support the conclusions?

Reviewer #1: Yes

Reviewer #2: Partly

2. Has the statistical analysis been performed appropriately and rigorously? 

Reviewer #1: N/A

Reviewer #2: Yes

3. Have the authors made all data underlying the findings in their manuscript fully available?

Reviewer #1: No

Reviewer #2: No

4. Is the manuscript presented in an intelligible fashion and written in standard English?

Reviewer #1: Yes

Reviewer #2: No

5. Review Comments to the Author

Reviewer #1: To

The Editor

Thank you for the opportunity to review this manuscript. The manuscript is well written and appreciate the hard work involved in the conduct of the study

Please see inputs section-wise below

Material and method

Suggest to follow the COREQ guideline http://www.ijo.in/documents/05COREQ SS.pdf

1. Please mention the research team and their credentials, who conducted qualitative interviews and FGDs, experience, and training.

2. Study design should include

• The theoretical framework used (methodological orientation) (e.g., grounded theory, discourse analysis, ethnography, phenomenology, content analysis)

• Participant selection: method of approach, reasons for refusals (if any), any experience/challenges in selecting participants for focus group discussion, how to overcome those challenges (can also be discussed in the discussion section)

• In the data collection section, mention any repeat interview (if any) and reason for the same. Please upload the Interview guide as supplementary material.

1. Data analysis: how the member checking was conducted (at present the sentence seems to be incomplete)

Result

Suggestion to reduce the quotable quotes or present the quotes in a tabular format theme-wise

Discussion

o Editing is required to avoid repetition from the result section

o What’s new in this study? Why should this article be cited? What is that ‘X’ factor?

o How the barriers (home visits) in Africa or your region is different from other low and middle-income countries?

o What significant steps were taken by the other African countries or low and middle-income countries to address barriers

o How the findings could be used: application/ translation and way forward. Recommendation to the government to address the barriers

Manuscript presented in an intelligible fashion: Yes, however text and language editing is required.

Thanks

Best wishes

Narendra K Arora

Reviewer #2: 1. The entire manuscript requires extensive English language copy editing.

2. I recommended you to reduce word count

3. Please revise the "Introduction" just by focused on your topic.

4. As you sometimes used the term HEW and sometimes CHW, are they interchangeable?

5. The topic should be clarified for "Barriers and facilitators of scheduled postnatal home visit"

6. Most of the sentences in "Study settings" are not mandatory, also not clear. And your justification in selecting the study settings does not much scientifically sound.

7. The sample size lack theoretical ground.

8. Some of the sentences in "Data analysis" section still talk about data collection.

9. The manuscript does not follow the recommended reporting guidelines for qualitative study.

10. In "Results" direct quotations are too difficult to understand.

11. Please provide clear discussion concentrated on your key findings.

12. Please sufficiently address the limitations of your study.

6. PLOS authors have the option to publish the peer review history of their article (what does this mean?). If published, this will include your full peer review and any attached files.

Reviewer #1: No

Reviewer #2: **Yes: **Mehammed Adem Getnet

---

## [Author Response · Author response to Decision Letter 0]

19 May 2021

Responses to Reviewers’

We wish to express our appreciation to the reviewers for their comments, which have helped us to improve our manuscript. We have thoroughly revised our manuscript titled “Mothers will be lucky if utmost receive a single scheduled postnatal home visit”: an exploratory qualitative study, Northern Ethiopia. We have incorporated and highlighted the changes in the revised manuscript based on the suggestions made by the editor and reviewers. 

Yemane Berhane Tesfau

Below follows a point-by-point response to the reviewers’ comments. 

Responses to comments from reviewer #1

Overall comment: The manuscript is well written and appreciates the hard work involved in the conduct of the study.

• Thank you very much for appreciating and acknowledging our work.

Comment 1: Please mention the research team and their credentials, who conducted qualitative interviews and FGDs, experience, and training.

Response: Thank you for pointing out this and we have incorporated it in the data collection section. 

Comment 2: Study design should include

Response: We agree with your comment and we have included:

• The methodological orientation: exploratory qualitative research using thematic analysis

• Participant selection: sampling, method of approach, and sample size 

-we have no any refusal to participation in the study

• In the data collection section mention any repeat interview: we did not have any repeat interview

Comment 3: Data analysis: how member checking was conducted

Response: we agree with you and we have re-written as:

To verify our interpretation as sound, we the authors checked the document and tried to brief about the contents to a sample of the participants in the study districts. 

Results section:

Comment 4: suggestion to reduce the quotable quotes 

Response: we agree with you and we tried to reduce and summarize the quotes.

Discussion 

Comment 5: Editing is required 

Response: we agree with your constructive comments and we did it.

Comment 6: What is new in this study? Why should this article be cited?

Response: Thank you for your constructive comments and we revised it to address these issues. Except for the sake of political agenda, attention was not given to postnatal care by the health care authorities in northern Ethiopia. The focus of healthcare authorities in northern Ethiopia were on other maternal and child health care services like coverage of 4th ANC, improving facility delivery, and child immunization; despite evidences of maternal and newborn deaths at home during the postnatal period.

Comment 7: How the barriers (home visits) in Africa or your region is different from other low and middle- income countries?

Response: Thank you we agree with you. And we included in the discussion section.

Comment 8: What significant steps were taken by the other African countries or low and middle-income countries to address barriers?

Response: Thank you for your insight. We included this in the introduction section of the manuscript.

Comment 9: How the findings could be used: application/translation and way forward. Recommendation to the government to address the barriers

Response: thank you for your constructive insight, and we incorporated in the conclusion part.

Responses to comments from Reviewer #2

Comment 1: The entire manuscript requires extensive English language copy editing.

Response: Thank you very much and we did it with language expert.

Comment 2: I recommended you to reduce word count

Response: Thank you and we agree with you, especially in the result section.

Comment 3: Please revise the “introduction” just by focused on your topic. 

Response: We absolutely agree with your concern and we tried to re-write focusing on the topic.

Comment 4: As you sometimes used the term HEW and sometimes CHW, are they interchangeable?

Response: Thank you for your observation, we can use interchangeable, however, in Ethiopia HEWs are government employed community health workers. In other countries the CHWs might be government employed or volunteers. There could be also difference in level of certification/education.

Comment 5: The topic should be clarified for “barriers and facilitators of scheduled postnatal home visit”

Response: Thank you for your suggestion and we appreciate your concern. The authors’ thinks scheduled PNHV in this study area does almost not exist. Thus, this topic expresses beyond barriers and facilitators so that healthcare authorities and concerned government bodies could give emphasis for its implementation.

Comment 6: Most of the sentences in “Study settings” are not mandatory, also not clear. And your justification in selecting the study setting does not much scientifically sound.

Response: Thank you very much and we agree with you.

Comment 7: The sample size lack theoretical ground.

Response: Thank you for your insight and were revised.

Comment 8: Some of the sentences in “data analysis” section still talk about data collection.

Response: yes it seems data collection but it is to mean about checking the data quality. Any way we revised it.

Comment 9: The manuscript does not follow the recommended reporting format for qualitative study.

Response: We have followed the PLOS ONE guideline Consolidated criteria for reporting qualitative research (COREQ), though there are different reporting formats for qualitative research

Comment 10: In “Results” direct quotations are too difficult to understand.

Response: Thank you for your comment and we have rephrased the quotes and tried to reduce the word counts

Comment 11: Please provide clear discussion concentrating on your findings.

Response: Thank you for your comment and agree with you.

Comment 12: Please sufficiently address the limitations of your study

Response: thank you and we addressed it.

We again thank the editors and reviewers for their insightful and constructive comments.

---

## [Editor Report · Decision Letter 1]

10 Jan 2022

PONE-D-20-24965R1“Mothers will be lucky if utmost receive a single scheduled postnatal home visit”: an exploratory qualitative study, Northern EthiopiaPLOS ONE

Dear Dr. Tesfau,

Thank you for submitting your manuscript to PLOS ONE. After careful consideration, we feel that it has merit but does not fully meet PLOS ONE’s publication criteria as it currently stands. Therefore, we invite you to submit a revised version of the manuscript that addresses the points raised during the review process.

We look forward to receiving your revised manuscript.

Kind regards,

Johnson Chun-Sing Cheung, D.S.W.

Academic Editor

PLOS ONE

Journal Requirements:

Additional Editor Comments (if provided):

Thank you for resubmitting this paper. I am aware that you have addressed to most of the reviewers' concerns already. However, an additional round of English copyediting work is required, for example in the Abstract section, it is quite awkward to mention "08 Key informant interviews", and "03 focus group" in the main text. Another critical issue is that given the number of participants is quite low, I am afraid some of the participants could be potentially identified by having their information provided in "Table 2" and also other parts of the paper. Please consider how could their privacy be protected in this regard.

---

## [Author Response · Author response to Decision Letter 1]

20 Feb 2022

Responses to Reviewers’ and Editors

We wish to express our appreciation to the reviewers for their comments, which have helped us to improve our manuscript. We have thoroughly revised our manuscript titled “Mothers will be lucky if utmost receive a single scheduled postnatal home visit”: an exploratory qualitative study, Northern Ethiopia. We have incorporated and highlighted the changes in the revised manuscript based on the suggestions made by the editor and reviewers. 

Yemane Berhane Tesfau

Below follows a point-by-point response to the editor’s and reviewers’ comments. 

Responses to editors

Comments: Please review your reference list to ensure that it is complete and correct. If you have cited papers that have been retracted, please include the rationale for doing so in the manuscript text, or remove these references and replace them with relevant current references. Any changes to the reference list should be mentioned in the rebuttal letter that accompanies your revised manuscript. If you need to cite a retracted article, indicate the article’s retracted status in the References list and also include a citation and full reference for the retraction notice.

Response: Thank you very much for the comments and concerns you raised about the reference list. We have not cited or removed retracted papers. In general we have not changed the reference list except we change the reference formatting to be in line with PLOS one referencing style.

Additional Editorial comments

Comments: Thank you for resubmitting this paper. I am aware that you have addressed to most of the reviewers' concerns already. However, an additional round of English copyediting work is required, for example in the Abstract section; it is quite awkward to mention "08 Key informant interviews", and"03 focus group" in the main text. Another critical issue is that given the number of participants is quite low, I am afraid some of the participants could be potentially identified by having their information provided in "Table 2" and also other parts of the paper. Please consider how their privacy could be protected in this regard.

Response: Thank you very much for appreciating and acknowledging our work. We duly agree with your constructive comments and we did it. For the additional round of English language copy editing we did it with language expert. We rephrased and incorporated the comments raised in the abstract section of the manuscript. The number of the participants was based on the saturation of the data (participants’ descriptions become repetitive).i.e. we continued sampling the participants until no new information emerged and saturation was reached. During designing the data collection, for example, we had planned to conduct four FGDs (two FGD from each district) in addition to the IDIs and KIIs, however, when no new idea was sought from the participants we conducted three FGDs. We have also included different participants to triangulate the data.

Regarding to the privacy of the participants: Thank you very much for your critical observation that some of the participants could be potentially identified by having their information provided in "Table 2”. We found that health extension program coordinators could be identified, because, in each district there is only one health extension program coordinator. Thus, we changed health extension program (HEP) coordinator to district level healthcare authority, as HEP coordinator is a member of district level healthcare authority. We also omitted some of the characteristics of the participants from table “Table 2”. Other participants could not identify (for example: health extension worker supervisors could not identified, because there are many health centers in each district).

In general, participants were de-identified throughout the transcription to ensure the confidentiality and anonymity of the participants. During audio recording, the file names were de-identified using codes and were deleted after transferring from recorder to personal protective device. Personal identifiers were also omitted during writing the description. Confidentiality was strictly maintained throughout the study and only the researchers have access to the data. No personal identifying information was retained that could identify the participants. Confidentiality was also practiced at every stage including anonymising the quotes from the interviews.

We again thank the editors and reviewers for their insightful and constructive comments.

---

## [Editor Report · Decision Letter 2]

1 Mar 2022

“Mothers will be lucky if utmost receive a single scheduled postnatal home visit”: an exploratory qualitative study, Northern Ethiopia

PONE-D-20-24965R2

Dear Dr. Tesfau,

We’re pleased to inform you that your manuscript has been judged scientifically suitable for publication and will be formally accepted for publication once it meets all outstanding technical requirements.

Kind regards,

Johnson Chun-Sing Cheung, D.S.W.

Section Editor

PLOS ONE

---

## [Editor Report · Acceptance letter]

21 Mar 2022

PONE-D-20-24965R2 

“Mothers will be lucky if utmost receive a single scheduled postnatal home visit”: an exploratory qualitative study, Northern Ethiopia 

Dear Dr. Tesfau:

I'm pleased to inform you that your manuscript has been deemed suitable for publication in PLOS ONE. Congratulations! Your manuscript is now with our production department. 

Kind regards, 

on behalf of

Dr. Johnson Chun-Sing Cheung 

Section Editor

PLOS ONE